# Understanding the Global Challenges to Accessing Appropriate Wheelchairs: Position Paper

**DOI:** 10.3390/ijerph18073338

**Published:** 2021-03-24

**Authors:** Rosemary Joan Gowran, Nathan Bray, Mary Goldberg, Paula Rushton, Marie Barhouche Abou Saab, David Constantine, Ritu Ghosh, Jonathan Pearlman

**Affiliations:** 1School of Allied Health, Faculty of Education and Health Sciences, Health Research Institute, Health Implementation Science and Technology (HIST), University of Limerick, V94 T9PX Limerick, Ireland; 2School of Health and Sport Science, University of the Sunshine Coast Locked Bag 4, Maroochydore, QLD 4558, Australia; 3Assisting Living and Learning Institute (ALL), Maynooth University, W23 VP22 Co. Kildare, Ireland; 4School of Health Sciences, Bangor University, Gwynedd LL57 2EF, UK; n.bray@bangor.ac.uk; 5Department of Rehabilitation Sciences & Technology, University of Pittsburgh & International Society of Wheelchair Professionals, Pittsburgh, PA 15206, USA; mgoldberg@pitt.edu (M.G.); jpearlman@pitt.edu (J.P.); 6École de Réadaptation, Université de Montréal, Montréal, QC H3C 3J7, Canada; paula.rushton@umontreal.ca; 7CHU Sainte Justine Research Centre, Montréal, QC H3T 1C5, Canada; 8Unité des Aides Technique, SESOBEL, Ain El Rihani, Lebanon; aidestechniques@sesobel.org; 9MBE, Founder Director, Motivation UK/International, Bristol BS3 4EG, UK; Constantine@motivation.org.uk; 10Mobility India, Rehabilitation Research and Training Centre, Rajiv Gandhi University of Health Sciences, Bengaluru 560078, India; ritugm@mobility-india.org; 11ICFAI University, Jharkhand 835222, India; 12International Society of Wheelchair Professionals, Pittsburgh, PA 15206, USA

**Keywords:** appropriate wheelchairs, assistive technology, human security, sustainable development

## Abstract

*Introduction:* Appropriate wheelchairs are often essential for the health and wellbeing of people with mobility impairments to enhance fundamental freedoms and equal opportunity. To date, provision has mainly focused on just delivering the wheelchair instead of following an evidence-based wheelchair service delivery process. In addition, many governments have not committed to a national wheelchair provision policy. *Approach:* To prepare this position paper, a systemic development model, founded on the sustainable human security paradigm, was employed to explore the global challenges to accessing appropriate wheelchairs. Positions: I: Consideration of key perspectives of wheelchair provision across the life course is essential to meet the needs to children, adults, older people and their families; II: Comprehensive wheelchair service delivery processes and a competent workforce are essential to ensure appropriate wheelchair service provision; III: Evaluations on wheelchair product quality development, performance and procurement standards are key as wheelchair product quality is generally poor; IV: Understanding the economic landscape when providing wheelchairs is critical. Wheelchair funding systems vary across jurisdictions; V: Establishing wheelchair provision policy is a key priority, as specific policy is limited globally. *Conclusion:* The vision is to take positive action to develop appropriate and sustainable wheelchair service provision systems globally, for me, for you, for us.

## 1. Introduction

Assistive products, such as wheelchairs, are an “essential component for inclusive sustainable development” [1]. Appropriate wheelchairs are often essential for the health and wellbeing of people with mobility impairments to enhance fundamental freedoms and equal opportunity. Every aspect of the wheelchair provision process will affect a person’s life positively or negatively depending on the experience. In 2008, the World Health Organization (WHO) published Guidelines on the provision of Manual Wheelchairs in Less Resourced Settings (LRS), which emphasized the need for appropriate wheelchairs, with a provision system addressing design, production, supply and service delivery processes [2]. Although the guidelines were directed toward LRS, they are relevant globally [3]. Some progress has been made; however, evidence suggests that despite the WHO’s guidelines on wheelchair provision, getting “the right wheelchair” (manual or electric-powered), in “the right way” [4], and learning how to use the device properly remains a global challenge. Provision has been focused around just delivering the product, instead of following an evidence-based service delivery process. In addition, many governments have not committed to national wheelchair provision policies.

Several authorities employ ad hoc, unsustainable systems instead of providing accessible person-centered services, skilled personnel, quality products, training, maintenance, follow up and management. To this end, universal wheelchair provision appears piecemeal, with tenuous links among stakeholders, such as wheelchair service users and families, therapists, service providers, manufacturers, regulators and policy makers. This creates systems that continue to be untenable, leaving people vulnerable to human insecurity [5,6,7]. Countries fail to provide appropriate and sustainable wheelchairs service delivery systems. These concerns were further reiterated in January 2018, when the United States Agency for International Development (USAID), World Learning and the International Society of Wheelchair Professionals (ISWP) facilitated a wheelchair stakeholders meeting hosted at Mobility India where fifty-six sector leaders shared perspectives and considered future developments. The meeting goal was to establish key priorities for the next five years to strengthen wheelchair services through policies, trained personnel and a range of appropriate wheelchairs. To achieve this goal, ten priority actions were identified to affect change towards sustainable development. These actions include: (1) building awareness by means of a global campaign involving key stakeholders; (2) conducting research which generates data to inform and strengthen appropriate wheelchair provision and practice; (3) establishing global service standards representative of service users, providers, educators and policy makers; (4) establishing product standards through evidence-based quality standards; (5) fostering innovation supporting user-centered design; (6) improving wheelchair supply, quality and procurement efficiency; (7) promoting evidence-based informed policy; (8) stimulating collaboration through multistakeholder platforms, supporting user-driven advocacy and champions; (9) supporting competency development to promote competent wheelchair sector personnel; and (10) supporting good practice through in-country initiatives, capacity building and policy development [8].

### Sustainable Human Security Perspective

The authors of this position paper believe that there is a need for sustainable wheelchair provision systems. Wheelchair provision communities of practice [5,7,9], working collaboratively to meet the Sustainable Development Goals (SDGs) [10] and adhering to the principles of the United Nations Convention on the Rights of Persons with Disabilities (UNCRPD), are key factors for a sustainable future [11]. A sustainable human security perspective enables an evaluation of the potential threats that inappropriate wheelchair provision has on protecting the health and welling being of individuals, families and communities. Human security is multifaceted and is defined in the General Assembly resolution 66/290 as “…an approach to assist Member States in identifying and addressing widespread and cross-cutting challenges to the survival, livelihood and dignity of their people” [12] Unstable wheelchair provision systems can lead to human security threats which may include poverty, unemployment, lack of basic healthcare and education, resources and access to appropriate products, personal and community inclusion and participation. Limited political will to create policy and regulate services confounds these security issues. An integrated approach is needed to develop wheelchair provision systems that are “people-centered comprehensive, context-specific and prevention-oriented responses that strengthen the protection and empowerment of all people” [12].

## 2. Approach

This position paper employs a systemic development model (SDM), founded on the sustainable human security paradigm, as defined by the World Engagement Institute (WEI) [13]. Four interlinked sustainability pillars (natural, social, economic and political) are included in the SDM to understand personal, organizational and institutional capacity: natural (health), social (culture), economic (technology) and political (law) aspects [13]. Readers are referred to the WEI website for further details on SDM. Thus, aligned with the SDM sustainability pillars, this position paper explores the following domains:The importance of appropriate wheelchairs to enhance the health, wellbeing and fundamental freedoms of people who require wheelchair services across the life course (social and natural);Context specific wheelchair service delivery systems and situations. (social, natural, economic, political);The economic landscape when providing products and services and the viability of provision (economic, political);The product production environment to access quality wheelchair and seating assistive technology (economic, natural);Political governance priorities to access appropriate wheelchairs (political, economic).

This exploration is supported by a search of the scientific and grey literature from 2008 to 2021, as well as case studies to highlight context specific examples.

## 3. Key Positions 

We propose five position points to address the global challenges to access appropriate wheelchairs towards sustainable wheelchair provision.

**Position** **I.**
*Consideration of key perspectives of wheelchair provision across the life course is essential to illustrate the multitude of variables affecting the transition to wheelchair use for children, adults and older people and their families. Indicators note the essentiality of the wheelchair as a lifeline, and the consequences of poor provision systems on quality of life, health and wellbeing must be considered when developing sustainable wheelchair provision systems.*


Wheelchair use does not discriminate; any person could need a wheelchair in their lifetime. The meaning of the wheelchair for people, children, adults and older people who require one is well documented, noting its essentiality as a lifeline and freedom to personal mobility and daily living across the life course [14,15,16]. Gowran et al. [17] when exploring wheelchair service user perspectives concluded that appropriate wheelchair provision “is a basic human right, supported by the essential and embodied nature of the wheelchair…” and noted that ad hoc services have created barriers to individual freedoms throughout life. Bray et al. [18] reported young children and parents’ perspectives on mobility-related quality of life are linked to “participation and positive experiences; self-worth and feeling fulfilled; and health and functioning”, and provision was noted by Labbé et al. [19] as a dynamic process when meeting the needs of older adults.

Wheelchair users and their families face many health and wellbeing challenges [20]. Additionally, acquiring an appropriate wheelchair is uniquely complex. The process is dependent upon personal narratives, reason for wheelchair use, diagnosis, life stage, secondary complications and living conditions across the globe [21,22].

For example, children and their families transitioning to wheelchair use should enable participation in play, school and community engagement and, as Casey et al. note, “reduce stress and burden” [23]. Wheelchair provision processes influence the health and wellbeing outcomes for the child and the caregivers [18,23,24,25,26]. Evidence suggests that while parents value the benefits of the wheelchair, they also have to continuously advocate for correct wheelchair prescription and follow-up services [24]. Wheelchair service provision for children and young people requires an anticipatory approach to re-examine for a child’s growth and developmental needs and be provided in a family-centered way and in tune with the emotional sensitivities experienced by caregivers [25]. The service also needs to account for growth and developmental needs. In addition, twenty-four-hour postural management should complement wheelchair education and training skills throughout life [27,28,29].

In adulthood, transition to wheelchair use is often difficult as a result of a sudden and traumatic event or condition such as spinal cord injury, brain injury and stroke or a progressive neurological condition. Having no prior knowledge or personal experience of wheelchair use pre-injury accelerates the need to adapt and adjust to the “unknown world of wheelchairs”, “navigating body changes” and “reshaping identity”. [30] One’s transition is influenced by the quality of the wheelchair provision experience [30,31,32]. People with spinal cord injury, as one example, are vulnerable to pressure injuries and numerous other secondary health complications which have significant treatment and management costs [33]. For instance, it is estimated that the National Health Service (NHS) in the United Kingdom (UK) spends between as much as GBP (British pounds) 2.6 billion on the treatment and management of pressure injuries each year [34]. Thus, a service that accounts for preventative care can largely influence the user’s outcomes.

To the contrary, evidence suggests that poorly prescribed and inappropriate provision has irreversible long-term effects causing incapacity to participate and vulnerability to early death [20]. In addition, untimely follow-up, maintenance and repair services, particularly when the wheelchair breaks down, have a negative impact on education and employment [35,36], increasing the risk of poverty and ill health. 

Other external variables impact a wheelchair user’s experience. When a user becomes a parent, they often experience social prejudice, personal and environmental obstacles. Additional challenges include those related to finance and support networks and easy access to assistive technology services [37].

As people age, health conditions increase and physical endurance declines, leading to dependence [38]. For older people, appropriate choice of a mobility device requires careful consideration to support independence in the home [39]. Acceptance of wheelchair use for personal mobility is difficult and further confounded by caregiver burden [40]. With aging populations, old age/nursing home accommodation is on the rise, with approximately eighty percent [41] of residents requiring wheelchairs. Access to appropriate wheelchair provision services is limited in many cases due to the complexity of the setting and high numbers of staff turnover [42]. This results in poorly fitted and maintained wheelchairs that are prescribed with limited postural support and inadequate pressure care management [41]. Individuals at the advanced stages of dementia are particularly vulnerable to pressure injury and postural deformity [43,44].

**Position** **II.**
*Comprehensive wheelchair service delivery processes and a competent workforce are essential to ensure appropriate wheelchair service provision to avoid navigation of poor service systems which generate stress and burden for the people accessing services and the personnel providing them. Building capacity and delivering adequate education and training for all is key to developing sustainable wheelchair provision systems.*


The WHO recommends an eight-step wheelchair service provision process (i.e., referral and appointment, assessment, prescription, funding and ordering, product preparation, fitting, user training and follow-up, maintenance and repairs). Evidence from several studies has described the positive impacts of this process on wheelchair user satisfaction, participation, health, quality of life, daily wheelchair use and activities of daily living [6,7,8]; however, this process is not consistently used in wheelchair service programs across low- to high-income countries. A factor of increasingly accepted importance is the shortage of competent personnel. In addition, wheelchair service users’ relationships with wheelchair service personnel reflect both positive and negative experiences [45]. There are emerging concerns in higher-income countries about a disconnection from person-centered practice affecting core philosophies, principles and ethical standards of practice, leaving wheelchair users feeling vulnerable and reluctant to complain about inappropriate provision and personnel susceptible to moral distress and burnout [46,47].

A key issue with how wheelchair services are provided is that across both low- and high-income settings, there are few countries that recognize a specific professional related to wheelchair service provision [6], likely impacting both the training that is provided and existing capacity. Several recent studies [48,49,50] have identified that there are disparities related to providers’ or students’ current knowledge of appropriate wheelchair service provision and the amount of wheelchair service provision training provided around the world. That adequate training is not included in health and social care training programs is reflective of the lack of specificity related to wheelchair education in most professional rehabilitation bodies’ educational standards, e.g., [51,52,53,54]. Fung et al. [48] identified across several contexts (e.g., region, income level, and type of training program) the amount of wheelchair training is highly variable, with programs citing anywhere from 2 to 40 h in a global survey. In a follow-up study, Fung et al. [49] identified several barriers, including limited funding, limited expertise, limited awareness of and training for instructors, limited physical resources (wheelchairs, related equipment, access to local clinics) and physical space limitations. As a result, student (and likely future provider) knowledge on the wheelchair service provision process is lacking. For example, Toro-Hernández investigated undergraduate physiotherapy (n = 2) [50] and occupational therapy (n = 7) [55] programs and determined that students’ (N = 199) knowledge does not align with the WHO eight-step wheelchair service provision process. These findings related to insufficient knowledge and capacity may suggest a commonality across other lower-income settings where training is often limited [6].

Novel training methods such as hybrid (i.e., part in-person, part online) methodology may help to offer adaptable and less costly alternatives to wheelchair service provision training [56]. In a recent study, both in-person and hybrid learning methodologies had a statistically significant effect on increasing wheelchair service knowledge with overall high levels of satisfaction, with the in-person group reporting overall larger effects when compared with the hybrid methodology [57]. Organizations may improve hybrid learning interventions based on best practices as recommended by Caulfleid [58] to enhance participants’ learning experiences and reduce potential barriers and limitations [58]. Ideally, these trainings will include several wheelchair sector stakeholders, including governments and multinational organizations. When delivered within an accompanying capacity building framework including training of trainers, an infrastructure can be developed to increase personnel competency. This will elevate the quality of services, awareness of product standards and the demand for comprehensive procurement systems. A recent example of such a program was facilitated by ISWP in the Dominican Republic [59]. A hybrid basic training and training of trainers program were sponsored by CONADIS, the National Council of Disability, with funding from the Pan-American Health Organization (PAHO). This style of training was particularly helpful during the COVID-19 pandemic, where 6 weeks of training were held online through asynchronous online modules and synchronous recitations to review modular content. The in-person sessions were reduced to 3 days due to the online content and were held with all participants and trainers adhering to WHO guidelines, including social distancing and mask-wearing measures.

Further, the use of open-source resources (e.g., WHO Wheelchair Service Training Package, Wheelchair Skills Program and platforms designed to share wheelchair education information, i.e., the Seating and Mobility Academic Resource Toolkit (SMART)) [60,61,62] may help to advance the training provided to future wheelchair service providers. However, despite the availability of these resources, their integration into existing or new academic and regional training programs can be a challenge dependent upon context-specific factors (e.g., time, knowledge and resources available) [47]. To guide instructors globally in the “how to” integrate open-source resources into training programs, an Educators’ Package will be an important next step [59]. A 30-person task force from 21 countries is collaborating to develop this package which will include the “how to” for advocacy, planning, course development, course implementation and course improvement specific to wheelchair provision education. The package will undergo external review and pilot testing, with subsequent refinements, prior to its launch in 2022. Promotion of open-source resources and how to integrate these resources into educational offerings through professional organizations’ communication channels (e.g., World Federation of Occupational Therapy (WFOT) Bulletin, ISWP Hub Newsletter), facilitation of dialogue amongst professionals via online forums (e.g., WFOT’s Occupational Therapy International Online Network and ISWP’s Wheelchair International Network) and continued work by groups dedicated to improving wheelchair education (e.g., ISWP’s Integration Committee, International Society of Prosthetics and Orthotics’(ISPO) Wheelchair Advisory Group) may also serve to enhance wheelchair education and capacity building of adequately educated professionals. In addition, national ministries globally could stipulate mandatory education qualifications in the wheelchair and seating product prescription and delivery. This is demonstrated by New Zealand’s national training for wheeled mobility and postural management credential responding to the accreditation framework 2010, to ensure proper assessment of ministry of health funded products [63,64].

**Position** **III.**
*Evaluations on wheelchair product quality development, performance and procurement standards are key as wheelchair product quality is generally poor and there is a need to strengthen performance measures and processes for design, testing and procurement.*


The WHO’s consensus definition of an appropriate wheelchair is “a wheelchair that meets the user’s needs and environmental conditions; provides proper fit and postural support; is safe and durable; is available in the country; and can be obtained and maintained and services sustained in the country at the most economical and affordable price” [1]. This underlies the importance of developing and utilizing reliable manufacturing and performance standards and procurement procedures to maximize the likelihood that users have access to appropriate wheelchairs, and that over time, the supply of inappropriate wheelchairs is eliminated.

Unfortunately, there is convincing evidence that wheelchair product quality is generally poor and has not improved over time, highlighting the importance of strengthening performance measures and procurement processes. Evidence that poor-quality, inappropriate wheelchairs were being widely distributed in less-resourced countries [1,65,66,67] can be combined with recent systematic research findings in higher resourced environments [68,69,70] that more than one half of all wheelchair users experience a breakdown every six months. The two key factors that are associated with frequent breakdowns are lack of routine maintenance and design shortcomings. For instance, a study of 95 wheelchair users found that lack of routine maintenance was associated with a 10-fold increase in the likelihood of a wheelchair breakdown [71]. Similarly, in a randomized control trial, a wheelchair maintenance intervention with 216 wheelchair users found that maintenance significantly reduced breakdown incidence [72]. To support preventative maintenance, training programs have been developed for both in-person and remote training and proven effective in increasing the ability for service providers, wheelchair users and their caregivers to perform preventative maintenance [72,73]. While these efforts have been fruitful in developing standardized training packages that can support and help to promote routine maintenance, if the wheelchair is poorly designed, it will break down, sometimes catastrophically, regardless of the maintenance performed. 

Assurance of a quality design can be done by evaluating the performance of wheelchairs through standardized testing methods during the design process, as well as once a wheelchair is in production. Standardized tests have been developed to measure wheelchair quality, including those presented in Table 1. In spite of the fact that these are formally developed international consensus standards, only a small minority of countries require wheelchairs to meet these standards prior to import or sale. From our perspective, this is a major reason that poor-quality wheelchairs are pervasive globally, and national adoption of product standards through a standardized procurement process is essential to address the problem.

Product procurement is the process of selecting and purchasing products. This could occur at several scales. For instance, a wheelchair service may procure products to provide choice to their clients. At a larger scale, procurement can happen at a national or provincial level. We recommend large-scale purchasing similar to what would occur at the national or provincial level in all instances. The same principles apply for procuring at a smaller scale in that they will provide a more systematic approach to screen out poor-quality products. A critical aspect of procurement is that products are appropriate for the client population, namely, they are readily available, are low-cost, repairable locally with access to spare parts and meet the user’s needs in their environment. 

Assuring appropriate products requires expertise in contracting, clinical service provision and the technical aspects of wheelchairs. As a single person rarely has all of these skills, purchasing committees are often established to support procurement. 

Recommended steps for appropriate procurement include:Performing a situational analysis to determine the range and quality of products that will be needed for the population;Developing a tender that includes key performance requirements such as independent standardized test results;Opening and marketing the tender to attract a large number of suppliers to respond to increase the product options; andPerforming a thorough unbiased technical product review carried out by a broad range of personnel with a range of technical and clinical experience.

Readers are referred to “*Design Considerations for Wheelchairs Used in Adverse Conditions*”, an expansive document describing best practices for design, testing and procurement, which was developed by a team of technical experts coordinated by the International Society of Wheelchair professionals (ISWP) [74]. To promote and advocate for wheelchair testing, ISWP has developed an open-source wiki which provides detailed test methods, designs of testing equipment and locations of test-labs globally (Available online: https://wheelchairnetwork.org/wheelchair-testing/; Accessed on 22 March 2021). 

**Position** **IV.**
*Understanding the economic landscape when providing wheelchairs is critical. Wheelchair funding is either through government or charity with systems varied across jurisdictions. Restricted access to wheelchairs creates a cycle of poverty and disability. Governments need to invest in systems to enable universal health coverage and, where possible, transition to local manufacturing of wheelchairs to produce sustainable, long-term services which support wheelchair users to take part in all aspects of life.*


The cost of even basic wheelchairs can be beyond the means of many individuals, particularly in low- and middle-income countries. Without government or charity support, many people with impaired mobility do not have essential mobility aids. For instance, “lack of economic means” is a key factor in restricted access to assistive technology in low-income countries such as Bangladesh [75]. Even in high-income countries, wheelchair users are forced to purchase equipment privately; it is estimated that in the UK, parents regularly self-fund 85% of the cost of powered wheelchairs for their children due to a lack of NHS coverage [76]. The situation is not helped by high import duties and informal charges levied on medical appliances in many low-income countries, such as Ghana [77].

Restricted access to wheelchairs creates a cycle of poverty and disability: “the disability causes poverty, and the poverty increases their disability” [78]. Microfinancing can help individuals to afford wheelchairs through personal loans. Daher et al. [78] found that this was a favorable approach for individuals with impaired mobility in Syria, but this still places the financial burden on the individual. Universal coverage of wheelchair provision should therefore be a goal for all nations, to enable people to live with dignity and to escape the cycle of poverty.

WHO succinctly identifies three models of wheelchair product provision [79]:Importation of complete wheelchairs;Importation of wheelchair components for local assembly;Local manufacturing of wheelchairs.

Wheelchair provision in low-income countries has historically been based on either donation or importation of complete wheelchairs, with price as the key selection criterion. This often leads to issues with service efficiency and wheelchair quality. For instance, in Tajikistan, wheelchairs are typically of low quality and inconsistent with international quality standards, leading to wheelchairs lasting little more than a year of regular use [79]. The low quantity of importation (around 800 wheelchairs per year) in Tajikistan causes long waiting lists for government-provided wheelchairs. Furthermore, limited resources to support essential services associated with wheelchair provision, such as maintenance and repair, have further compounded the issues of inefficient and unreliable wheelchair provision. In line with WHO guidelines [1], Tajikistan is now moving toward a service-provision model of wheelchair distribution, with an aim to provide universal coverage of wheelchairs by 2023. 

Significant government commitment is required to ensure that large-scale contracts and national procurement strategies are in place to support universal coverage of wheelchairs. Low-income countries, such as Tajikistan, need to first increase capacity to assemble wheelchairs locally and, in the long-term, develop infrastructure to manufacture wheelchairs using locally sourced materials. Significant net benefits can be achieved by moving toward local production of wheelchairs, including increased employment and manufacturing skills [79]. One approach to reducing the cost of wheelchair provision is to promote sustainable practices through refurbishment and recycling. It is estimated that 50% of all wheelchairs supplied by the NHS are refurbished [80]; cost savings of between 9% and 14% are achieved through this approach [81]. At present, many wheelchair users pay for their own repair and maintenance, particularly in low-income countries, which impacts quality standards and affordability.

In order to appropriately allocate government resources to wheelchair provision, it is essential that the full spectrum of costs associated with wheelchair provision be identified. A number of financial factors must be taken into account, including the capital cost of equipment, customization and environmental adaptation; staff costs; service overheads; repair and maintenance costs; and training. Lack of economic evidence is still a major hurdle in improving wheelchair provision; thus, future research must focus on developing a better understanding of what approaches to wheelchair procurement and provision work in different contexts.

The UNCRPD states that assistive technology is essential to enable people to be independent, to participate in all aspects of life and to exercise their personal rights [82]. Furthermore, the UNCRPD emphasizes the importance of personal mobility and equal access to assistive technology to facilitate the highest degree of independence for each individual. Without adequate wheelchair provision, many people are caught in a cycle of poverty and deprivation and consequently have reduced access to education, work and social facilities [1]. People with disabilities are more likely to be unemployed compared to non-disabled peers and when employed tend to earn less [83]. These issues also have national economic impacts due to loss of productivity and health service resource use (World Health Organization, 2011). Therefore, universal health coverage for wheelchair provision is needed to promote equal access to wheelchairs. However, across the world, most people who require a wheelchair do not have access to one [1,83]. This is a consequence of limited availability of wheelchairs, high cost of product, lack of awareness and a paucity of trained professionals to facilitate provision, particularly in low-income countries. Governments need to invest in infrastructure to enable universal health coverage and, where possible, transition to local manufacture of wheelchairs to produce sustainable, long-term services which support wheelchair users to take part in all aspects of life.

In order to understand the efficiency of different ways of providing wheelchairs, it is important to measure and evaluate both the costs of different provision models and also the subsequent health, wellbeing and financial impacts to wheelchair users. Determining the incremental costs and benefits of alternative provision models will support governments to maximize health outcomes for wheelchair users. For instance, if certain wheelchair interventions are not found to be cost-effective, then alternative approaches could be prioritized. The routine application of methods of health economics could support service-commissioning and funding allocation decisions and enable wheelchair provision practices to be evidence-based and equitable. This first requires the development of robust methods of outcome measurement within this context.

In order to examine the relationship between costs and outcomes associated with medical interventions, health economists commonly undertake economic evaluations. In recent evidence syntheses, there was found to be little robust economic evidence regarding pediatric wheelchair provision [84,85]. This is in part due to the difficulties of applying traditional methods of economic evaluation in the context of wheelchair provision, where patient groups and outcomes are often less homogeneous than other clinical areas.

Future research should focus on developing robust evidence of cost effectiveness to guide wheelchair service development and intervention provision, including the development of novel approaches to cost and outcome measurement.

**Position** **V.**
*Establishing wheelchair provision policy is a key priority, as specific policy is limited globally. There is a call for governments to commit to developing and implementing national sustainable wheelchair provision strategies.*


A significant amount of research across the globe has been conducted, evaluating wheelchair service delivery systems, and evidence suggests that while there is some funding, albeit limited, for wheelchair products, getting the right wheelchair is challenging. In many cases, the overall efficiency of wheelchair provision processes is poor [5,7,86,87,88] despite the availability of guidelines for good practice [2,83,84]. While the Norwegian service delivery system appears to be the most visible exemplar for appropriate assistive technology provision, specifically related model wheelchair service delivery systems are not easily identified [89,90]. Based on the available evidence, the recommendations lead to a call for government commitment to develop and implement national strategies, yet specific government legislation and policy as to the appropriate provision of wheelchairs and ongoing support to enhance the health and wellbeing for people requiring wheelchair services is not visible or easily accessible. In addition, there is little or no evidence that evaluates good practice to provide a blueprint or template for countries to follow to support sustainable development of wheelchair service infrastructure [91,92,93,94,95,96].

Situational analyses capture context-specific conditions for wheelchair provision (for example, Ireland, Romania, the Philippines and Tajikistan [5,7,79], with the view to developing and implementing strategic plans by engaging with key stakeholders, including governments, in conversation for change and long-term commitment to sustainable service provision. Contexts reviewed are diverse, with distinct geographic, demographic and sociopolitical governance dimensions and require individualized strategic planning. Contextual diversity reflects the type of wheelchair services available across countries from low to high income. Nonetheless, many of the key issues affecting appropriate provision are similar to assume responsibility for oversight of the provision system. Common components include the need for advocacy, wheelchair service infrastructure, product and procurement standards, capacity building, education, training and further research. The World Health Organization (WHO) is taking steps to meet these needs, announcing on 3 December 2020 the development of the Global Standards for Wheelchair Service Provision. Key international stakeholders will produce, review and approve this universal document, making it relevant across all income settings. Application of these standards will play a pivotal role when addressing global challenges to access appropriate wheelchairs set out in this position paper. In addition, the publication of the Assistive Products Specifications (APS) global guidebook for quality manufacturing 2 March 2021 presents details of twenty-six prioritized assistive products to support improved procurement of appropriate wheelchairs and other assistive technologies [97]. The starting point within each country to address wheelchair provision challenges will differ, and successful application of standards and guidelines should be based on comprehensive situational analysis findings and strategic planning.

## 4. Conclusions

Universal consciousness and commitment to change in wheelchair service provision that meets peoples’ needs now and in the future is the key. Evidence indicates the importance of understanding “across the life course experiences” as essential to lifelong access to appropriate wheelchair services. 

Demand, need, availability and supply of appropriate wheelchairs is complex. Studies highlight a lack of consistency in how services are prioritized and regulated, with a dearth of appropriately trained personnel, resulting in poorly delivered services affecting peoples’ lives. In addition, funding and the lack of policies that guide (and require) appropriate wheelchair provision are the primary challenges that restrict access. Therefore, conveying the complexity of wheelchair provision in context to meet peoples’ health and wellbeing needs across the life course is challenging. Understanding in-country perspectives and inclusive solutions, reflecting specific personal, social, economic, environmental, historical and political nuances, which connect with the priorities of national governments is important. 

In order to take action, the wheelchair provision experience globally needs to be reviewed and reflected upon, informing the wheelchair sector on next steps to strengthen evidence-based, adequately resourced, integrated wheelchair services supported by policies, competent personnel and a range of appropriate products. There is a need to build self-sustaining networks and best practice in wheelchair service provision around the world to bridge the gap to access appropriate wheelchairs for all at all ages from a sustainable human security perspective. The vision is to take positive action to develop appropriate and sustainable wheelchair service provision systems globally, for me, for you, for us.

## Figures and Tables

**Table 1 ijerph-18-03338-t001:** Standardized tests.

Name	Scope
ISO 7176 https://www.iso.org/committee/53792/x/catalogue/ (accessed on 22 March 2021)	Covers testing for manual and power wheelchairs and is used as a reference for most national standards
Whirlwind ISO+ https://whirlwindwheelchair.org/simplified-strength-testing-of-manual-wheelchairs/ (accessed on 22 March 2021)	Low-cost approach to ISO 7176, with additional tests focused on conditions in LREs
ISO 16480 https://www.iso.org/committee/53792/x/catalogue/ (accessed on 22 March 2021)	Test for Wheelchair seating Systems
WC 19 (accessed on 22 March 2021)	Wheelchair Crash Testing
ISWP Wheelchair Standards (accessed on 22 March 2021)	Caster, rolling resistance, corrosion, and whole-wheelchair testing

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
