# Peer review of "Understanding the Global Challenges to Accessing Appropriate Wheelchairs: Position Paper"

_ijerph, 2021, doi:10.3390/ijerph18073338_

Round 1
Reviewer 1 Report
As per the first content of my review, the authors seem to have modified the abstract, Key position and references, and they seem to have modified the entire sentence to make it easier to read. Therefore, this study is evaluated in terms of journal publication conditions.
Reviewer 2 Report
This revised version of the manuscript is very clearly written. The answers to the comments are satisfactory. And the article has considerable gained in legibility. It is clearly a position paper and the authors are thoroughly involved in their domain. A lot of literature is cited to reinforce their pleading.
This manuscript is a resubmission of an earlier submission. The following is a list of the peer review reports and author responses from that submission.
Round 1
Reviewer 1 Report
In my opinion, this article adds very little to existing knowledge. It provides only a review of existing literature to support its two main theses: 1) Appropriate wheelchair and seating assistive technology is essential for the health and wellbeing of people with mobility impairments, to enhance fundamental freedoms and equality of opportunity; 2) countries fail to provide appropriate and sustainable wheelchairs service delivery systems. Moreover, the second of these theses seems to me insufficiently supported, because only very few specific country-studies have been revised, most of them from quite small countries.
No empirical research –surveys, interviews with stake holders, legislative or policy review- has been undertaken by the authors in addition to already existing literature.
Therefore, although the research topic is in my opinion extremely interesting and convenient, I think that the present paper seems to be a political declaration rather than a scientific article.
It is necessary to add to the literature review new empirical research. In that case we would have new evidence supporting the theses of the authors and giving an stronger support to policy recommendations.
Author Response
Notes:
- We thank the Reviewers for their thoughtful comments.
- We have numbered the comments.
- Line numbers referred to in the Responses are those of the revised manuscript with changes shown.
- Changes in the manuscript are in red, as we have made major revisions as request and many tracked changes, which we accepted for clarity.
REVIEWER #1
Reviewers’ Comment #1: In my opinion, this article adds very little to existing knowledge. It provides only a review of existing literature to support its two main theses: 1) Appropriate wheelchair and seating assistive technology is essential for the health and wellbeing of people with mobility impairments, to enhance fundamental freedoms and equality of opportunity; 2) countries fail to provide appropriate and sustainable wheelchairs service delivery systems. Moreover, the second of these theses seems to me insufficiently supported, because only very few specific country-studies have been revised, most of them from quite small countries.
Response: The 5 proposed key positions limiting sustainable wheelchair service provision systems are now more clearly described in section 3 in red. You are correct, there is limited literature on this topic and we have reported the situational analyses that we have found to exist.
Reviewers’ Comment #2: No empirical research –surveys, interviews with stake holders, legislative or policy review- has been undertaken by the authors in addition to already existing literature. Therefore, although the research topic is in my opinion extremely interesting and convenient, I think that the present paper seems to be a political declaration rather than a scientific article.
Response: Thank you for this observation. The intention as you have rightly pointed out is one of position rather than a scientific article. We have adjusted the language in lines 86-106 and 108-109 to enhance the clarity of our position for this position paper.
Reviewers’ Comment #3: It is necessary to add to the literature review new empirical research. In that case we would have new evidence supporting the theses of the authors and giving an stronger support to policy recommendations.
Response: It was not our intention to provide new evidence, only to share our position and support it with literature and our combined experiences. Our understanding of a position paper is that it requires three parts: background information, evidence to support our position and a conclusion. Unlike a research paper, a position paper does not require new evidence.
Reviewer 2 Report
Some of the authors are engaged for many years in the research on utilization of wheelchairs. The subject of the article is clear and important. But by reading the content I was a little embarrassed. The text is rather long and with exceptionally long phrases which make it difficult to read. The text has a lot of similarities of a literature review, but it isn't. At the other side, it is not a scientific research paper either. In the section on approach, the authors said they were using a mixed-method approach. But the methods used are not described in detail. Which quantitative methods are used? And which qualitive methods? Focus groups, mind mapping, .....??? Which in-country case studies are used?
The utilization of the SDM concept, developed by WEI, was not clearly described or adapted to the access and utilization of wheelchairs. Findings were presented in another format and did not have a clear relationship with personal, organisational of institutional capacity.
In other words, authors did not clearly make a choice:
A position paper, without a scientific connotation, can be extremely useful. In that case the text can be shortened by 50%. But the real message can be more clearly presented.
In a literature review, the methodology of the selection of the literature must be thoroughly described, objective by objective. And their will be less possibilities to make political statements or to advocate for better access of wheelchairs. The building of awareness cannot be an objective but just a result of publishing the article.
Author Response
Notes:
- We thank the Reviewers for their thoughtful comments.
- We have numbered the comments.
- Line numbers referred to in the Responses are those of the revised manuscript with changes shown.
- Changes in the manuscript are in red, as we have made major revisions as request and many tracked changes, which we accepted for clarity.
REVIEWER # 2
Reviewers’ Comment #1: Some of the authors are engaged for many years in the research on utilization of wheelchairs. The subject of the article is clear and important.
Response: Thank you. No change needed.
Reviewers’ Comment #2: But by reading the content I was a little embarrassed. The text is rather long and with exceptionally long phrases which make it difficult to read.
Response: Thank you. We have edited the text throughout the entire manuscript to improve its readability. Because of the substantial revision we did not use highlight in red for all of these changes.
Reviewers’ Comment #3: The text has a lot of similarities of a literature review, but it isn't. At the other side, it is not a scientific research paper either. In the section on approach, the authors said they were using a mixed-method approach. But the methods used are not described in detail. Which quantitative methods are used? And which qualitive methods? Focus groups, mind mapping, .....??? Which in-country case studies are used?
Response: Thank you. You are correct. We have modified lines 86 to 106 in the Approach section to more accurately reflect our methods.
Reviewers’ Comment #4: The utilization of the SDM concept, developed by WEI, was not clearly described or adapted to the access and utilization of wheelchairs. Findings were presented in another format and did not have a clear relationship with personal, organisational of institutional capacity.
Response: Thank you. We have more explicitly linked our proposed key positions with the 4 sustainability pillars in lines 91 to 103.
Reviewers’ Comment #5: In other words, authors did not clearly make a choice: A position paper, without a scientific connotation, can be extremely useful. In that case the text can be shortened by 50%. But the real message can be more clearly presented. In a literature review, the methodology of the selection of the literature must be thoroughly described, objective by objective. And their will be less possibilities to make political statements or to advocate for better access of wheelchairs. The building of awareness cannot be an objective but just a result of publishing the article.
Response: Thank you. We have adjusted the language in lines 86 to 87, 108 to 109 and stated each position point to enhance the clarity that this is a position paper. We have significantly reduced the amount of text throughout the paper.
Reviewer 3 Report
I think this study is very meaningful as we are discussing issues and solutions that have recently become issues using position paper on global issues for proper wheelchair access. Thank you for evaluating the content of these meaningful issues. However, the author needs to modify with a few amendments.

Author Response
Notes:
- We thank the Reviewers for their thoughtful comments.
- We have numbered the comments.
- Line numbers referred to in the Responses are those of the revised manuscript with changes shown.
- Changes in the manuscript are in red, as we have made major revisions as request and many tracked changes, which we accepted for clarity.
REVIEWER # 3
Reviewers’ Comment #1: Keywords- Any of the keywords that are too common should be deleted.
Response: Thank you. In lines 34-35 we have modified the key words.
Reviewers’ Comment #2: Introduction - In the introduction of the study, it is necessary to add a general description of the necessity of a wheelchair, who is the specific target group required, and the current problem of wheelchairs (if possible).
Response: Thank you. We have included this information in Postion I, lines 165-222.
Reviewers’ Comment #3: In the finding of this paper, it will be easier to understand if it is divided into two main categories: all problems related to wheelchairs, and solutions to problems.
Response: Thank you for your suggestion. As we have made major revisions to clearly presented this work as a position paper, we have presented the problems and solutions within each position point. We hope this is satisfactory.
Reviewers’ Comment #4: In addition, subtitles for each area should be classified more clearly and explained in a directional way. The description of the current title is a bit complicated, repetitive and little bit confusing.
Response: Thank you for this observation. We have modified the subtitles to be Position statements.
Reviewers’ Comment #5: References - You will need to reorganize the references to fit the format of the journal submission.
Response: Thank you. The majority of references now conform to the journal’s guidelines. We will ask the editorial committee for advice regarding some references highlighted in yellow.
Round 2
Reviewer 1 Report
I think the paper has been substantially improved, and I accept the comments of the authors, in their response to my previous report, clarifying that this is a position paper and not a research article.